# Calcium-Deficiency during Pregnancy Affects Insulin Resistance in Offspring

**DOI:** 10.3390/ijms22137008

**Published:** 2021-06-29

**Authors:** Junji Takaya

**Affiliations:** Department of Pediatrics, Kawachi General Hospital, 1-31 Yokomakura, Higashi-Osaka 578-0954, Osaka, Japan; takaya@kawati.or.jp; Tel.: +81-72-965-0731; Fax: +81-72-965-2022

**Keywords:** calcium, glucocorticoid receptor, homeostasis model assessment of insulin resistance, homeostasis model assessment of beta-cell function, insulin resistance, pregnancy, 11β-hydroxysteroid dehydrogenase-1

## Abstract

Prenatal malnutrition is known to affect the phenotype of the offspring through changes in epigenetic regulation. Growing evidence suggests that epigenetics is one of the mechanisms by which nutrients and minerals affect metabolic traits. Although the perinatal period is the time of highest phenotypic plasticity, which contributes largely to developmental programming, there is evidence of nutritional influence on epigenetic regulation during adulthood. Calcium (Ca) plays an important role in the pathogenesis of insulin resistance syndrome. Cortisol, the most important glucocorticoid, is considered to lead to insulin resistance and metabolic syndrome. 11β-hydroxysteroid dehydrogenase-1 is a key enzyme that catalyzes the intracellular conversion of cortisone to physiologically active cortisol. This brief review aims to identify the effects of Ca deficiency during pregnancy and/or lactation on insulin resistance in the offspring. Those findings demonstrate that maternal Ca deficiency during pregnancy may affect the epigenetic regulation of gene expression and thereby induce different metabolic phenotypes. We aim to address the need for Ca during pregnancy and propose the scaling-up of clinical and public health approaches that improved pregnancy outcomes.

## 1. Introduction

Adverse nutritional conditions during pregnancy may permanently alter the structure or function of specific organs in the offspring, leading to various chronic diseases in adulthood [1]. When the intrauterine environment is nutritionally restrictive, growth is impaired initially to conserve nutrients for the fetal brain and, under more severe conditions, to protect the mother. Although such changes may be protective at first, they can become irreversible, and the offspring are left to cope, even in adulthood, with the various adaptations that were made in response to nutritional restriction as a growth-impaired fetus. This is the basis of the thrifty phenotype model originally proposed by Hales and Barker [2] and now referred to as the Developmental Origins of Health and Disease (DOHaD) [3].

Maternal undernutrition, and the consequent low birth weight of offspring, predisposes the offspring to various diseases, including adult-onset insulin resistance syndrome [4,5]. Dietary protein restriction in pregnant rats results in offspring with hypertension, hyperglycemia, altered hepatic enzyme profiles, or a combination of these conditions [6]. Deficient calcium (Ca) intake in pregnant rats results in increased blood pressure in the offspring during adulthood [7].

This review highlights the evidence for the impact of Ca-deficient programming on insulin resistance in offspring and the role of nutritional interventions as a reprogramming strategy in the emerging area of DOHaD research.

The aim of this narrative review was to assess scientific studies that evaluated the effects of Ca deficiency during pregnancy and/or lactation on insulin resistance in the offspring of animal models, and human clinical trials. Scientific studies published between 1990 and 2020 in the databases of PubMed and Cochrane databases were identified using specific search terms. Key search terms, including MeSH terms, were determined in accordance with the PICO method: “calcium deficiency”, “pregnancy”, “lactation”, and “epigenetics” for inputs and “fetus”, “offspring”, and “metabolic diseases” for outcomes. Publications that did not specifically quantify changes in maternal undernutrition were excluded, resulting in a total of 233 articles that were selected for preparing this review. The major findings relating to the Ca deficiency during pregnancy, including lactation, and epigenetics are discussed in this review.

## 2. Placental Transport of Calcium

Although recommended Ca intake in pregnancy is 1000 mg/day in both the EU and the United States, globally, most countries fall below this level [8,9]. Low Ca intake during pregnancy increases the risk of preeclampsia, gestational diabetes, and preterm birth [10,11,12]. Total serum Ca levels in normal pregnancy reduced, especially in the third trimester of pregnancy [13,14,15]. This has been explained by the increasing demands of the growing fetus [16]. Placental transportation of Ca is key for fetal bone development [17]. This Ca transport is dependent on vitamin D and Mg, which are also commonly insufficient during pregnancy [18,19]. For adequate bone development to occur, the human fetus requires approximately 30 g of Ca during development, which is actively transported across the placenta [20]. Maternal parathyroid hormone (PTH) does not cross the placenta but affects the fetus by altering the concentration of maternal circulating Ca. Placental Ca transfer is also regulated by parathyroid hormone-related protein (PTHrP). The syncytiotrophoblast layer of the human placenta transfers as much as 30 g of Ca from the mother to the fetus, especially in late gestation, where Ca transport through different channels must increase in response to the demands of accelerating bone mineralization of the fetus [21]. Placental Ca transport is dependent upon a series of transport proteins in the syncytiotrophoblast. A family of plasma membrane ATPase isoforms (PMCA1-4) are thought to be a rate-limiting step in the placental Ca transportation [22]. Insufficient Ca supply during fetal development also causes the future growth delay and risk of osteoporosis in the offspring. Significant maternal hypocalcemia impairs the delivery of Ca to the fetuses, who may develop secondary hyperparathyroidism, skeletal demineralization, and fractures [23].

## 3. Calcium and Insulin Resistance

Ca is an important second messenger in signal transduction pathways that regulate a wide variety of processes, including gene expression, protein synthesis, secretion, muscle contraction, metabolism, and apoptosis [24]. A link between Ca intake and insulin resistance in obesity and metabolic syndrome has been identified in epidemiological studies [25,26]. Several observational prospective studies have also shown a relationship between low or insufficient oral Ca intake and the incidence of type 2 diabetes mellitus (DM2) [27,28] and metabolic syndrome [29]. It was shown that individuals who have poor Ca intake present higher body weight [30]. Furthermore, a Ca-rich diet is known to improve insulin sensitivity [31,32]. A systematic review of randomized clinical trials suggested that Ca supplementation induces a small, but statistically significant, weight loss in overweight and obese individuals [33]. Earlier dose-dependent meta-analyses of cohort studies have shown that dietary intake of Ca prevents the development of DM2 [34,35]. Recently Wu et al. reported that in the large prospective cohort study, higher serum Ca levels precede peripheral insulin resistance, and this relation plays a role in the development of hypertension [36].

However, the mechanisms underlying this relationship remain poorly understood. Dietary Ca appears to play a pivotal role in the regulation of energy metabolism and obesity risk. Ca has the ability to modulate energy metabolism through calciotropic hormone concentrations: calcitriol and parathyroid hormone (PTH) [30]. A high-Ca diet is known to attenuate body fat accumulation and weight gain during periods of overconsumption of an energy-dense diet and promote fat breakdown and preserve metabolism during periods of caloric restriction, thereby markedly accelerating the loss of weight and fat [37]. Thus, a diet that is poor in Ca could inhibit lipolysis, stimulate lipogenesis, and decrease lipid oxidation [38]. Severe Ca deficiency increases visceral fat accumulation, down-regulating genes associated with fat oxidation, and increases insulin resistance while elevating serum PTH in estrogen-deficient rats [39]. The specific roles of Ca signaling and endoplasmic reticulum stress affect the development of insulin resistance and atherosclerosis [40].

Vitamin D plays a major role in Ca ion homeostasis by regulating Ca transport and bone mineralization. Vitamin D deficiency is associated with direct effects on offspring health such as low birth weight, poor skeletal health, obesity, and insulin resistance [41,42]. Similarly, prenatal vitamin D deficiency is associated with increased insulin resistance and inflammatory mediators in childhood [43,44]. A study of rats revealed that offspring who were born from mothers under vitamin D deficiency had increased fatty acid and markers of inflammation and oxidative stress in the liver and higher prevalence of liver steatosis [45]. In a mouse study, maternal vitamin D deficiency induced structural remodeling of the pancreas and impaired insulin secretion due to reduced gene expression of PDX-1, which regulates the expression of GLUT2, glucokinase, and insulin in adult offspring [42]. Maternal Vitamin D deficiency in a large birth cohort involving Indian children predicted higher insulin resistance at 9.5 years old [41].

## 4. Calcium and Epigenetics

The effects of the nutritional status of the mother have been discussed for many years, and several studies have considered the nutritional status of the mother during pregnancy as an environmental epigenetic factor that may play an important role in fetal development [46]. Regulatory regions of the genome can be modified through epigenetic processes during prenatal life. The modification of chromatin and DNA contributes to a large well-documented process known as “programming”. Programming of fetal insulin resistance was reported to be induced by intrauterine abnormal activation of inflammation, adipokines, and the endoplasmic reticulum stress [47]. The correlation between gut dysbiosis and metabolic disturbance has attracted attention. Li et al. reported that imbalance in maternal Ca intake promotes body weight gain in offspring, which may be mediated by calcium’s modulation on the gut microbiota and lipid metabolism [48].

Epigenetics is the study of mitotically heritable alterations in gene expression potential that are not caused by changes in DNA sequences [49]. Epigenetic mechanisms, which are established during prenatal and early postnatal development, function throughout the lifetime of complex organisms to maintain the diverse gene expression patterns of different cell types. Several molecular mechanisms, including the methylation of cytosines within CpG dinucleotides, various modifications of the histone proteins that package DNA in the nucleus, and cell-autonomous expression of a myriad of auto-regulatory DNA-binding proteins, interact to perpetuate the regional chromatin conformation that dictates which genes will be transcriptionally competent in specific cell types [50]. Ca has been indirectly associated with epigenetic modifications [51]. Conjugated linoleic acid and Ca supplementation modified the methylation pattern of fatty-acid-related genes under a high-fat diet in adult mice [52].

## 5. 11β-Hydroxysteroid Dehydrogenase

### 5.1. Glucocorticoid and 11β-Hydroxysteroid Dehydrogenase-1

Preliminary data suggest that circulating cortisol concentrations are higher in patients with metabolic syndrome compared to healthy subjects [53,54,55]. Dysregulation of glucocorticoid action has been proposed to be one of the central features of metabolic syndrome [56]. In the major metabolic organs, tissue sensitivity and exposure to glucocorticoids are determined by the levels of intracellular peroxisome proliferator-activated receptor α (PPARα), glucocorticoid receptor (GR), and the activity of the microsomal enzyme 11β-hydroxysteroid dehydrogenase-1 (11β-HSD1). 11β-HSD1 converts inactive glucocorticoids (cortisone in humans and 11-dehydrocorticosterone in rodents) to their active forms (cortisol and corticosterone, respectively) [57]. 11β-HSD1 is highly expressed in liver and adipose tissue, where glucocorticoids reduce insulin sensitivity and action [57,58,59]. The activity of 11β-HSD1 in liver and adipose tissue might contribute to the development of several features of insulin resistance or metabolic syndrome [60,61,62]. Obese individuals have increased *11β-HSD1* mRNA in both subcutaneous and visceral fat tissue [63]. Experimental studies have shown higher *11β-HSD1* expression in adipose tissue associated with features of metabolic syndrome such as increased waist circumference and insulin resistance [64].

Harno et al. reported that liver-specific *11β-HSD1* knockout mice given low-dose 11-dehydrocorticosterone do not show any of the adverse metabolic effects seen in wild-type mice [65]. This result implies that liver-derived intra-tissue glucocorticoids, rather than circulating glucocorticoids, contribute to the development of metabolic syndrome and suggest that local action within hepatic tissue mediates these effects. In contrast, Morgan et al. reported that adipose-specific *11β-HSD1* knockout mice given higher dose glucocorticoids are protected from hepatic steatosis and circulating fatty acid excess, whereas liver-specific *11β-HSD1* knockout mice develop full metabolic syndrome phenotypes [66]. This result demonstrates that 11β-HSD1, particularly in adipose tissue, is key to the development of the adverse metabolic profile associated with circulating glucocorticoid excess.

11β-hydroxysteroid dehydrogenase-2 (11β-HSD2) converts excess cortisol into inactive cortisone [67]. Phosphoenolpyruvate carboxykinase (PEPCK), a key hepatic gluconeogenic enzyme, simultaneously decarboxylates and phosphorylates oxaloacetate into phosphoenolpyruvate in one of the earliest rate-limiting steps of gluconeogenesis. 11β-HSD1 regulates key hepatic gluconeogenic enzymes, including PEPCK, through the amplification of GR-mediated tissue glucocorticoid action [57,59,67].

### 5.2. A Calcium-Deficient Diet Affects Hepatic 11β-Hydroxysteroid Dehydrogenase-1 Expression in the Liver of Dams

We previously reported that after 2 weeks of the low-Ca diet (low-Ca group: 0.008% Ca) or control diet (control group: 0.90% Ca), no differences in serum glucose, corticosterone, or insulin levels were observed between the two groups. In adulthood, 1 rat month is comparable to 3 human years [53]. The homeostasis model assessment of insulin resistance (HOMA-IR) has proved to be a robust tool for the assessment of insulin resistance [68]. The low-Ca group rats showed higher values of HOMA-IR (*p* < 0.05) and intact parathyroid hormone (*p* < 0.05) and lower values of adiponectin (*p* < 0.01). In the low-Ca group, the expression of hepatic *Hsd11b1* mRNA was up-regulated, and hepatic *Pck1* expression was down-regulated (*p* < 0.001). The expression levels of *Nr3c1*, *Ppara,* and *Hsd11b2* showed a similar tendency. The 2-week Ca-deficient diet in rats was associated with the upregulation of the hepatic expression of *Hsd11b1* mRNA, which occurred before the animals developed obesity or overt features of metabolic syndrome [69]. Over-activity of 11β-HSD1 is associated with increased intracellular active glucocorticoids [57,58,59]. Rodent genetic studies have suggested that increased *Hsd11b1* expression or activity increases the risk of several components of metabolic syndrome [60,61]. In summary, a low-Ca diet alters glucocorticoid metabolism, which leads to hepatic upregulation of *Hsd11b1*, and is possibly a key mechanism of the induction of metabolic complications caused by Ca deficiency [69].

## 6. A Ca-Deficient Diet in Pregnant or Nursing Rats Affects the Offspring

Lillycrop et al. reported that pregnant rats on a protein-restricted diet developed hypomethylation and increased expression from the *Ppara* and *Nr3c1* promoters in the liver of the offspring [70,71]. This demonstrates that maternal nutrition during pregnancy can affect the regulation of non-imprinted genes via the altered epigenetic regulation of gene expression, thereby inducing different metabolic phenotypes. A high-fat diet during pregnancy was reported to induce neonatal gender-specific hepatic fat accumulation by increased *pck1* expression and histone modification [72].

### 6.1. The Methylation of Specific Cytosines within the 11β-Hydroxysteroid Dehydrogenase-1 Promoter in the Liver of the Offspring

We investigated the methylation of individual CpG dinucleotides in glucocorticoid-related genes in liver tissue of neonatal offspring from Ca-deficient rat dams. Female rats consumed either a Ca-deficient (0.008% Ca) or control (0.90% Ca) diet ad libitum from 3 weeks before conception to 21 days after parturition. Pups were allowed to nurse from their original mothers and were then sacrificed on day 21. The methylation of CpG dinucleotides in the *Pck1* [73], *Ppara,*
*Nr3c1*, *Hsd11b1*, and *Hsd11b2* promoters was measured in liver tissue by pyrosequencing [65]. The methylation levels of all genes did not differ between groups, except for *Hsd11b1*, which was significantly lower in the rats from the Ca-deficient dams (*p* < 0.05). Serum corticosterone levels were higher in the male pups from the Ca-deficient dams than in those from the control dams (*p* < 0.05). The expression levels of *Pck1* and *Nr3c1* were significantly lower in the Ca-deficient group than in the control group, whereas those of *Hsd11b1, Hsd11b2,* and *Ppara* did not differ significantly [74].

Although the hepatic expression of *Hsd11b1* may have been initially up-regulated by epigenetic mechanisms in the offspring from Ca-deficient dams, *Hsd11b1* was likely down-regulated by other mechanisms during the early postnatal period. The methylation level of hepatic *Hsd11b1* was altered in the offspring as a consequence of the maternal dietary manipulation, but the epigenetic changes were not reflected in corresponding alterations in transcription. The nuclear receptor co-repressor complex is affected by environmental factors such as nutrients and hormones, which can lead to altered DNA methylation, acetylation, histone modification, other epigenetic changes, or some combination thereof; such epigenetic changes can and do alter the activity of DNA. These factors can also alter feedback loops involving nuclear receptors that normally regulate repression and maintain balance [75].

The down-regulation of *Hsd11b1* suggests that a compensatory mechanism may diminish cortisol production in the liver. Reduced hepatic glucocorticoid exposure also represents a compensatory mechanism that limits the metabolic complications of insulin resistance. In our study, no significant difference in serum 11β-HSD1 levels was found among the offspring groups; however, this may have been due to tissue-specific differences between serum and liver. Whether glucocorticoids modulate *Hsd11b1* expression is unknown, and *Hsd11b1* expression differs greatly between the liver and other tissues [76,77,78]. Obese rodents exhibit tissue-specific dysregulation of 11β-HSD1; it is usually up-regulated in adipose tissue and down-regulated in the liver [79,80]. In both obese Zucker rats and obese humans, 11β-HSD1 activity is high in adipose tissue but low in the liver [77,78,81]. In adipose tissue and smooth muscle cells, glucocorticoid induces *Hsd11b1* mRNA expression, but contradictory results have been obtained in the liver [78,81].

In summary, a Ca-deficient diet during pregnancy and nursing induced hypomethylation of specific CpG dinucleotides in the *Hsd11b1* promoter in the liver tissue of neonatal offspring. These changes in *Hsd11b1* expression likely contribute to marked increases in glucocorticoid hormone action in liver tissue [67] and potentiate the induction of insulin resistance during adult life [56].

### 6.2. A Ca-Deficient Diet in Dams during Gestation Increases Insulin Resistance in Male Offspring

The offspring rats of the same experimental methods as described in the previous section were raised to adults. Pups were allowed to nurse from their original mothers until weaning, when they were fed a control diet. The offspring were then sacrificed at an age of 180 days. The mean levels of insulin and glucose as well as the HOMA-IR values were higher only in the male offspring from the Ca-deficient dams than in those from the control dams (*p* < 0.01) [82]. In all offspring, the serum leptin levels were correlated with the serum insulin levels, and they were inversely correlated with the levels of ionized Ca.

A Ca-deficient diet in dams during gestation and early nursing may alter the glucocorticoid metabolism of her offspring, resulting in higher intracellular glucocorticoid concentration in the hepatic cells of the offspring; this abnormal glucocorticoid metabolism may induce the metabolic complications associated with Ca deficiency. Dietary Ca restriction in dams during pregnancy alters postnatal growth, the expression of *Hsd11b1*, and insulin resistance in a sex-specific manner.

### 6.3. Osteocalcin in the Offspring from a Ca-Deficient Dams

Osteocalcin (OC), or bone γ carboxyglutamic acid (Gla) protein, is the most abundant non-collagenous bone matrix protein [83]. OC is specifically expressed in osteoblast lineage cells and secreted from bone into the bloodstream [84]. OC is subjected to post-translational carboxylation by a vitamin K-dependent carboxylase to yield carboxylated (Gla-OC) and undercarboxylated (Glu-OC) molecules [85]. Glu-OC acts directly on pancreatic β-cells to increase insulin secretion, as well as insulin sensitivity and glucose tolerance [86,87,88]. The offspring rats of the same experimental methods as described in the previous section were raised to adults [82]. The mean levels of Glu-OC in Ca-deficient female offspring were higher than those in control female offspring and control male offspring. The mean levels of Gla-OC were higher in Ca-deficient female offspring than those in control female offspring, whereas no significant difference was observed in these measures between the two groups in male offspring. The effects of Glu-OC on glucose homeostasis have been reported to differ by sex [89]. Increased Glu-OC could contribute to lower insulin resistance in female Ca-deficient offspring, and therefore might be beneficial for glucose metabolism. Consequently, only male Ca-deficient offspring may acquire insulin resistance.

## 7. Interventions as Reprogramming Strategies

### 7.1. Mismatch Cross-Fostering

A tendency toward insulin resistance and its associated diseases are the potential maladaptive consequences of responses made in early life that either lead to immediate responses requiring fetal adaptations for survival (e.g., growth impairment) or, through anticipatory processes, induce a phenotype matched for a more thrifty adult environment [90]. At least in its anticipatory form, this is a normative physiological process [91] that can be considered the outcome of a mismatch between the phenotype defined by evolutionary and developmental processes and the energy environment the individual currently lives in [92].

Several reports have suggested that, at least in animal models, the developmental programming of adult diseases is potentially reversible by nutritional interventions during the period of developmental plasticity. Reprogramming strategies to reverse the programed development and achieve normal development may include nutritional intervention, exercise, lifestyle modification, or pharmacological therapy.

Next, we determined the effects of nursing in addition to Ca deficiency during pregnancy on hepatic *Hsd11b1* expression in the offspring. Female rats were fed either a Ca-deficient or control diet from 3 weeks before conception to 21 days after parturition. On postnatal day 1, pups were cross-fostered to the same or opposite dams and divided into the following four groups: CC, DD, CD, and DC, with the first letter indicating the diet of the original mother and the second letter indicating the diet of the nursing mother (C: control diet; D: Ca-deficient diet). All offspring were fed a control diet beginning at weaning and were sacrificed on day 200 [93].

In male offspring, the mean levels of insulin and glucose, as well as the HOMA-IR values [68], were higher in the DD and DC groups than in the CC group. We found no difference in HOMA-IR values between the CC and CD groups in both males and females. The expression of *Hsd11b1* was lower in male DD rats than in male CC rats (Figure 1). In the comparisons of CC vs. CD and DD vs. DC, *Hsd11b1* expression was higher in the male offspring nursed by cross-fostered dams than in those nursed by dams fed the same diet. There was a significant effect of sex (*p* < 0.0001) on 11β-HSD1 mRNA expression, which was lower in females. In females, *Hsd11b1* expression was higher in DC rats than in CC rats (Figure 1). Mismatched nutrition after birth, i.e., cross-fostered nursing, disrupted the adaptation that had been programmed in the fetus. If the nutritional environment established before birth is mismatched after birth, the balance established by the compensatory action of developmental programming responses made in early life collapses. Metabolic programming effects translate into an adverse intrauterine environment during pregnancy. An altered mismatched nutritional experience during the suckling period can impact adult health in the offspring [94,95].

### 7.2. Calcium Supplement

Oliveria et al. reported maternal nicotine exposure programs for higher central obesity and insulin resistance [96]. Nevertheless, Ca supplementation in reversing some endocrine–metabolic changes, such as central adiposity, leptin, and insulin resistance, which have been detected in adult offspring whose mothers were exposed to nicotine during lactation [97]. The further studies from the same team reported that early weaning leads to obesity, hyperleptinaemia, and insulin resistance in a rat model. However, dietary Ca supplementation seems to protect against the development of endocrine and metabolic disorders in the early weaning offspring through vitamin D inhibition [98]. Another team reported that maternal vitamin D supplementation increases maternal and infant 25(OH)D concentrations and significantly decreases maternal HOMA-IR and increases infant birth weight [99].

## 8. Sex-Specific Epigenetic Phenotypes

Male offspring (F1) developed insulin resistance after nursing from Ca-deficient dams. Maternal Ca restriction during pregnancy alters postnatal growth, *Hsd11b1* expression, and insulin resistance in a sex-specific manner [93]. Sex differences in metabolism are well established, and females have been shown to be more insulin-sensitive and secrete more insulin than males [100,101]. Previous reports have shown that the epigenetic effect induced by intrauterine growth restriction is gender-specific [102,103,104,105]. Hall et al. reported sex differences in genome-wide and gene-specific DNA methylation patterns that were associated with altered gene expression, microRNA levels, and insulin secretion in human pancreatic islets [106]. Testosterone may affect epigenetic regulation [107]; however, further investigation is necessary. In our study, we found pronounced sexual dimorphism in the levels of serum-ionized Ca, with higher levels in females (*p* < 0.05). Pronounced sexual dimorphism was also seen in the 11β-HSD1 levels in the liver, with lower levels in females [108].

Several reports have suggested sex-specific epigenetic phenotypes in animal studies. Chang et al. reported that maternal low Ca intake has effects on brain DHA accretion sex-specifically through epigenetic modification on fatty acid desaturase [109]. Li et al. reported that the effects of maternal Ca status on lipid metabolism were found only in the female adult offspring but were similar between offspring males and females at postnatal 21 days [110,111,112]. de Almeida Faria et al. reported that day-restricted, i.e., out-of-phase feeding during pregnancy and lactation leads to glucose intolerance in only male offspring, which is caused by a disruption in insulin secretion capacity. They also reported that this metabolic programming is possibly caused by mechanisms dependent on miRNA modulation of syntaxin 1a [113].

Ca supplementation of pregnant women with low calcium intakes altered the childhood trajectories of growth and bone and body composition development of their offspring in a sex-specific manner, resulting in slower growth among females compared to placebo and accelerated growth among males by age 8–12 years [114]. Calcium carbonate supplementation of pregnant mothers resulted in higher IGF1 in boys and lower IGF1 in girls during mid-childhood [115]. Clinically, there are reports of gender differences in insulin resistance during puberty and menopause but no reports of gender differences during the neonatal period.

This further supports the hypothesis that early postnatal lactation plays a sexually divergent role in the programming of phenotypes that appear later in life. To uncover the detailed mechanisms underlying the epigenetic modulation of genes due to different maternal nutritional insults, further studies are needed in diverse models of developmental programming.

## 9. Inheritance of Epigenetic Traits: Epigenetic Change in Three Generations of Offspring

Epigenetic models for the amplification of the prevalence of insulin resistance across generations do not necessarily involve transgenerational epigenetic inheritance. Maternal (F0) insulin resistance may induce somatic epigenetic alterations in her offspring (F1: first generation) that confer susceptibility to insulin resistance. Consequently, F1 females would have an increased likelihood of having insulin resistance in adulthood, resulting in the perpetuation or amplification of the insulin resistance epigenetic alterations in the F2 (second generation). The following rat studies provide evidence for this model.

Dearden and Balthasar reported that the increased female vulnerability for developing metabolic disturbances in response to the diet of the mother during pregnancy propagates a vicious cycle of obesity and DM2 in subsequent generations [116]. We investigated whether alterations in insulin resistance and secretion induced in F1 offspring by feeding dams (F0) a Ca-deficient diet during pregnancy and lactation are passed on with or without further change to two subsequent generations. Twelve-week-old virgin rats (F0) were fed with either a control (C) diet or a Ca-deficient (D) diet ad libitum for 3 weeks. Three weeks later, the rats were mated with males (F0 generation) who were fed with the control diet. F0 dams continued to consume their respective diets throughout gestation and lactation. F1 and F2 females were mated with males of each generation on postnatal day 70. F1 and F2 dams were fed with the control diet during pregnancy and lactation, and the offspring were weaned onto the control diet.

The homeostasis model assessment of beta-cell function (HOMA-β %), a surrogate estimate of pancreatic β-cell function based on the measurements of fasting plasma glucose and insulin concentrations, is routinely used to assess insulin secretory function [117,118]. The HOMA-β % value was decreased in the F1 through the F3 offspring of Ca-deficient F0 dams, and male F1 offspring from these dams developed insulin resistance. However, the insulin resistance recovered in F2 male offspring, and the HOMA-IR value was significantly lower and the serum ionized Ca was higher in the F3 offspring. The HOMA-IR was affected only in the F1 generation of males, and it was corrected after the F2. Although insulin secretion basically worsened, insulin resistance did not develop in the F2 and F3 offspring because of sex-specific modifications [119]. Our findings indicated that although the insulin resistance appeared to have been corrected, the ability to secrete insulin decreased with each passing generation. A mechanism to correct the tendency towards insulin resistance could function throughout the generations. The surrounding environment and adaptation by compensation might be involved. The intergenerational transmission of developmentally programmed insulin resistance is determined by the relative insulin sensitivity of the mother during pregnancy and/or lactation [120]. Benyshek et al. reported that insulin sensitivity was normalized in the F3 offspring of developmentally programmed insulin-resistant (F2) rats fed an energy-restricted diet [120]. Ca-deficiency in F0 pregnant rats did not consistently lead to insulin resistance in offspring over three consecutive generations. A gestating F0 female can generate F1 embryos and an F2 germline that were directly exposed to an environmental factor, and F3 becomes the first generation that was not directly exposed to the factor [105,121,122].

In summary, maternal Ca restriction during pregnancy alters postnatal insulin secretion for three generations and also sex-specifically affects postnatal insulin resistance.

## 10. Conclusions

Females have a much higher prevalence of inadequate Ca intake and would benefit the most from increasing dairy intake [123,124]. Maternal Ca deficiency during pregnancy can affect the epigenetic regulation of gene expression and thereby induce different metabolic phenotypes, such as insulin resistance, in their offspring (Figure 2). The epigenome is an important target of environmental modification. The developmental programming of insulin resistance is potentially reversible by nutritional interventions during periods of developmental plasticity.

## Figures and Tables

**Figure 1 ijms-22-07008-f001:**
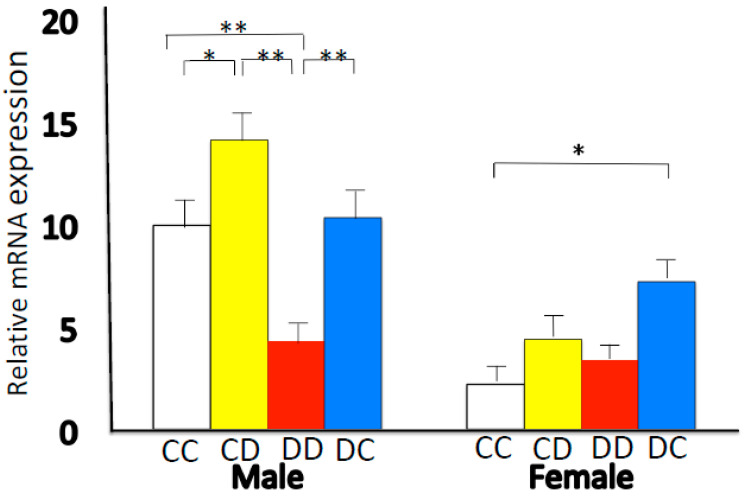
Hepatic mRNA expression levels of *Hsd11b1.* After delivery, foster mothers fed the same or different diets: CC, DD, CD, and DC (first letter: diet of original mother; second letter: diet of nursing mother. C: control diet; D; calcium-deficient diet). Data are represented as the means ± standard error. * *p* < 0.05, ** *p* < 0.01. This figure is quoted from Takaya J et al. *PLoS ONE* **2014**, *9*, e84125 [93].

**Figure 2 ijms-22-07008-f002:**
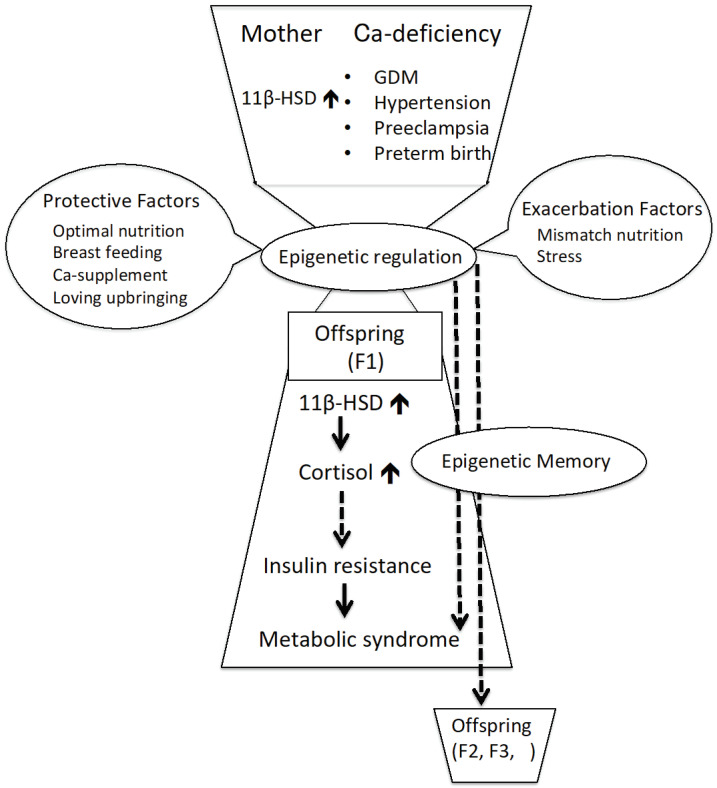
Maternal calcium deficiency and fetal programming with metabolic outcomes. Ca: calcium; GDM: gestational diabetes mellitus; 11β-HSD1: 11β-hydroxysteroid dehydrogenase-1; F1: first-generation offspring; F2: second-generation offspring; F3: third-generation offspring.

## Data Availability

Not applicable.

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
