# Peer review of "Calcium-Deficiency during Pregnancy Affects Insulin Resistance in Offspring"

_ijms, 2021, doi:10.3390/ijms22137008_

Round 1

Reviewer 1 Report

This review article summarizes the negative effects of Ca deficiency during pregnancy on the offspring and induction of different metabolic phenotypes. The role of Ca in fetal programing and correlation with metabolic disturbance is presented. The article is well written and thoroughly summarizes relatively recent findings. There are a few things that need to be addressed:

Point 1. To improve understanding of the topic, the author must provide a figure summarizing role of Ca and fetal programming with metabolic outcomes. I suggest a form of a graphical abstract.

Point 2.     I suggest to the title of the chapter 4 to better address the topic in both subchapters.  

Point 3. Dietary Ca restriction in dams during pregnancy alters postnatal growth, the expression of Hsd11b1, and insulin resistance in a sex-specific manner (Chapter 5). How can this be explained? Please provide some form of an explanation/hypothesis.  Is this found in human pregnancies? Is there any sex-specific correlation in Ca effects on insulin sensitivity, resistance, or metabolic phenotypes?

Point 4. I suggest adding one chapter on different epigenetic modifications and regulations, before the chapter 3.

 Point 5. In the experiments combining nursing and Ca deficiency (Takaya et al 2014), how the sex difference can be explained?

Point 6. In Conclusions, the author should mention the sex differences in dietary Ca deficiency. It should also contain some conclusion remarks for human pregnancy to reach a better pregnancy outcome, and Ca in diet as a possible improvement of human health.

Author Response

Thank you very much for your valuable comments and criticisms concerning our manuscript. We have carefully reviewed your comments and have made the point-by-point corrections described below.

  • Provide a figure summarizing role of Ca and fetal programming with metabolic outcomes

Response: Thank you for your instructive comments. Following the reviewer's advice, I added figure 2 summarizing role of Ca and fetal programing with metabolic outcomes.

2) Titles of the chapter 4 (Originally)

Response: Thank you for the constructive comments. I changed the title of the original chapter 4 (in reviced version Chapter 5) to better address the topic in both subchapter.

  • Sex-specific manner

Response: According to the Reviewer’s comments, we also referred the references(Ref #93, 100-115), and discussed and compared the results and contents of the papaers with our results (Chapter 8).

  • Adding one chapter on different epigenetic modifications and regulations

Response: Thank you for your constructive comments. However, many epigenetics reviews have been published so far, and I thought it was not necessary to add another chapter.

  • Sex difference in the experiments combining nursing and Ca deficiency

Response: Thank you for your constructive comments. Maternal Ca restriction during pregnancy alters postnatal growth, 11b-HSD1 expression, and insulin resistance in a sex-specific manner. We found a significant effect of sex (P < 0.0001) on 11b-HSD1 mRNA expression, which was lower in females. Although there was a sex difference in the expression level of 11b-HSD1, there was no sex difference in the tendency due to the difference in nursing, and both male and female offspring showed the same tendency. I added an explanation about sex differences in the text.

6) Sex difference in dietary Ca deficiency

Response: Thank you for your constructive comments. I mention that female have a much higher prevalence of inadequate Ca intake and would benefit the most from increasing dairy intake referring another references [Ref 123,124]. In conclusions, I remark that maternal Ca deficiency during pregnancy can affect the epigenetic regulation of gene expression, and thereby induce different metabolic phenotypes, such as insulin resistance, in their offspring.

Reviewer 2 Report

In this paper, the author reviews the evidence linking calcium-deficiency during pregnancy with development of insulin resistance in the offspring. The author proposes that the two events may be connected via sex-specific dysregulation of 11β-hydroxysteroid dehydrogenase-1 (Hsd11b1) expression in the liver via alterations in DNA methylation. Overall, I think there is some disconnection between the title and abstract on one side, and the content of the paper on the other.

Significant weak points of this review that should be addressed are the following:

  1. The key role that the author seems to imply for 11β-HSD1 in connecting calcium-deficiency to development of insulin resistance is not mentioned in the abstract, but makes the bulk of the review. The author should address this point by rewording the abstract.
  2. Given the topic of this paper, I think the author should review the mechanisms that are implicated in the transfer of calcium, across placenta, from the mother to the developing fetus.
  3. Calcium has a key role in bone development and function. The bone has been highlighted in recent years as a key player in regulating metabolism. The author focusses almost exclusively on an isolated example of gene dysregulation in the liver. However, according to the title of the paper, a more detailed discussion on the role played by other key organs is obviously missing.
  4. The link between calcium metabolism and DNA methylation is very tenuous. The author reviews some observations based on correlations, but does not make any significant attempt to provide explanations or even to speculate on the molecular mechanisms that could link calcium deficiency to altered DNA methylation patterns.

Minor point:

  • Phrase at lines 79-81 does not have a verb.

Author Response

Thank you very much for your valuable comments and criticisms concerning our manuscript. We have carefully reviewed your comments and have made the point-by-point corrections described below.

1) 11β-HSD1 in connecting calcium-deficiency to development of insulin resistance should be addressed in the abstract

Response: Thank you for your instructive comments. Following the reviewer's advice, I described the relationships of 11β-HSD1 related to insulin resistance in the abstract.

2) Review the transfer of Ca across placenta from the mother to the developing fetus.

Response: Thank you for the constructive comments. According to the Reviewer’s comments, I review the mechanisms that are implicated in the transfer of Ca across placenta from the mother to the developing fetus in the newly added Chapter 2.

3) A more detailed discussion on the role in bone in regulating metabolism

Response: Thank you for your instructive comments. As the reviewers pointed, bone can act as an endocrine organ through the secretion of bone-specific hormones, which are known as osteokines. At least two osteokines are implicated in the control of glucose metabolism: lipocalin-2 (LCN2) and osteocalcin (OC).

Added about osteocalcin in our experimental data in Chapter 6.3.

There have been numerous research reports on maternal calcium and fetal skeletal formation. However, DOHaD theory based on maternal undernutrition has been widely tested, but as far as we know, there are no papers that have studied the relationship between maternal hypocalcemia and ostiokine from the perspective of epigenetics. Future research is expected to clarify this correlation.

4) The link between calcium metabolism and DNA methylation

Response: We were able to find a phenomenon in which calcium deficiency alters the DNA methylation of certain genes, but unfortunately the mechanism is still unknown. Elucidation by future research is awaited.

Minor point:

Phrase at lines 79-81: I corrected the phrase.

Reviewer 3 Report

The review aims to identify the effects of calcium deficiency during pregnancy and/or lactation on insulin resistance in the offspring. 

I have a few revisions that the authors should make prior to publication. 

I think the authors should include some clinical data on the prevalence of calcium deficiency in both pregnant and non-pregnant individuals.

I also think the authors should include a summary figure that outlines how calcium impacts insulin resistance. 

The authors should also include search criteria for including articles in the review. 

Author Response

Thank you very much for your valuable comments and criticisms concerning our manuscript. We have carefully reviewed your comments and have made the point-by-point corrections described below.

1) Clinical data on the prevalence of calcium deficiency in both pregnant and non-pregnant individuals

       Response: Thank you for your instructive comments. Quann et al reported that females (ages 4-71 years) have a much higher prevalence of inadequate calcium intake [Ref #124]. Calcium deficiency is also apparent in non-pregnant women, and calcium deficiency is even more serious in pregnant women who need calcium for their fetal growth. As far as I know, I couldn't find any literature reports comparing non-pregnant women with pregnant women.

2) Summary figure that outlines how calcium impacts insulin resistance

Response: Thank you for your instructive comments. Following the reviewer's advice, I added a figure summarizing role of Ca and fetal programing with metabolic outcomes.

3) Search criteria for including articles in the review

Response: Thank you for your constructive comments. I include search criteria for including articles in the introduction part.

Round 2

Reviewer 1 Report

The authors sufficiently answered all my concerns. The Figure 2 is not properly displayed. Please make sure that the image is correctly displayed. 

Author Response

Response to Reviewer 1

Thank you very much for your valuable comments and criticisms concerning our manuscript. We have carefully reviewed your comments and have made the point-by-point corrections described below.

  • Figure 2 is not properly displayed

Response: Thank you for your kind comments. There is no problem with the original drawing submitted by us. There may be a system problem in converting to PDF for reviewers. I will attach it separately just in case.

Reviewer 2 Report

The revised version of this review took into account most of the recommendations I've made on the initial submission. I think the paper is overall in better shape. However, the author acknowledges that the links between calcium metabolism and epigenetic mechanisms of gene regulation remain largely unknown. In the abstract, the author states that "Those findings
demonstrate that maternal Ca deficiency during pregnancy affects the epigenetic regulation of gene expression...". This is an obvious overstatement and needs to be toned down. 

Minor points:

  • The newly added phrase at lines 71-72 does not read well.
  • Figure 2 looks distorted upon conversion into PDF.

Author Response

Reply to Reviewer #2

Thank you very much for your valuable comments and criticisms concerning our manuscript. We have carefully reviewed your comments and have made the point-by-point corrections described below.

1) Overstatement and needs to be toned down

Response: Thank you for your instructive comments. Following the reviewer's advice, I described the sentence to be toned down in the abstract.

  • The newly added phrase at lines 71-72

Response: I rewrite the phrase at lines 71-72.

  • Figure 2 looks distorted upon conversion into PDF

Response: Thank you for your kind comments. There is no problem with the original drawing submitted by us. There may be a system problem in converting to PDF for reviewers. I will attach it separately just in case.
